# Thermal and Energy-Efficiency Assessment of Hybrid CLT–glass Façade Elements

**Vlatka Rajčić [1],*, Nikola Perković [1], Chiara Bedon [2]**  **, Jure Barbalić [1] and Roko Žarnić [3]**

[1]   Structural Department, Faculty of Civil Engineering, University of Zagreb, 10000 Zagreb, Croatia; nperkovic@grad.hr (N.P.); jbarbalic@grad.hr (J.B.)

[2]   Department of Engineering and Architecture, University of Trieste, 34127 Trieste, Italy; chiara.bedon@dia.units.it

[3]   Faculty of Civil and Geodetic Engineering, University of Ljubljana, 1000 Ljubljana, Slovenia; roko.zarnic@fgg.uni-lj.si

*   Correspondence: vrajcic@grad.hr

**Abstract:** Façade elements are a building component that satisfies multiple performance parameters. Among other things, "advanced façades" take advantage of hybrid solutions, such as assembling laminated materials. In addition to the enhanced mechanical properties that are typical of optimally composed hybrid structural components, these systems are energy-efficient, durable, and offer lighting comfort and optimal thermal performance, an example of which is the structural solution developed in collaboration with the University of Zagreb and the University of Ljubljana within the Croatian Science Foundation VETROLIGNUM project. The design concept involves the mechanical interaction of timber and glass load-bearing members without sealing or bonding the glass-to-timber surfaces. Following earlier research efforts devoted to the structural analysis and optimization of thus-assembled hybrid Cross-Laminated Timber (CLT)-glass façade elements, in this paper, special focus is given to a thermal and energy performance investigation under ordinary operational conditions. A simplified numerical model representative of a full-size building is first presented by taking advantage of continuous ambient records from a Live-Lab mock-up facility in Zagreb. Afterwards, a more detailed Finite Element (FE) numerical analysis is carried out at the component level to further explore the potential of CLT–glass façade elements. The collected numerical results show that CLT–glass composite panels can offer stable and promising thermal performance for façades similar to national and European standard requirements.

**Keywords:** Cross-Laminated Timber (CLT); laminated glass; hybrid façade element; thermal performance; energy efficiency; numerical modelling

## 1. Introduction

In the last decade, the building construction sector has shown an increased use of load-bearing building members composed of timber and glass. The first applications and research efforts (see [1–4]) were focused on composite beams. Later, the attention of researchers progressively moved towards the definition of novel timber–glass solutions for curtain walls and enclosures to cover large surfaces and also offer enhanced load-bearing performance in buildings [5,6]. In most cases, the need for resistance and ductility under both ordinary design loads and extreme events like earthquakes typically resulted in design applications characterized by the use of a continuous (adhesive and/or mechanical) connection between a given timber frame and glass infill panels. Typical examples of these cellular frame-supported glass panels can be found in Figure 1.

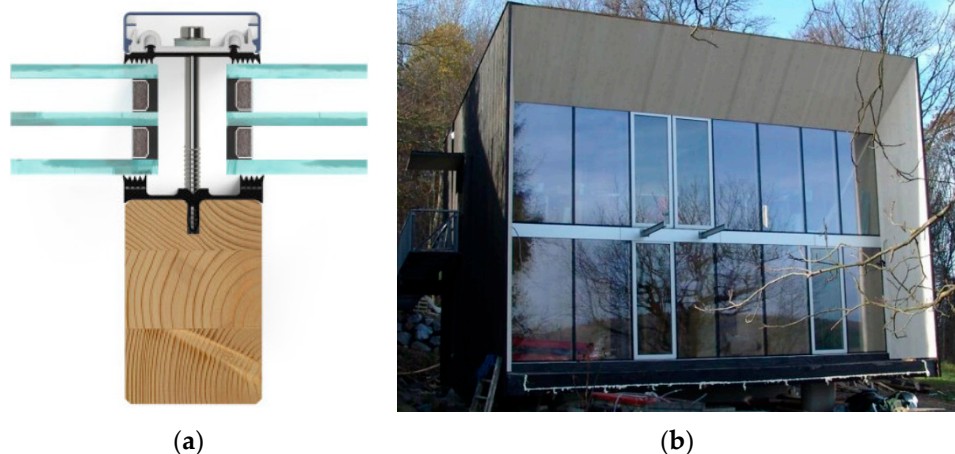

**Figure 1.** Examples of timber–glass façade solutions: (**a**) a typical cross-section of curtain wall detail (*www.stabalux.com*); (**b**) full-size, adhesively bonded framed panels (from [5]).

An innovative and promising solution (see Figure 2a and [7–9]) was found in the form of a hybrid, full-size, multi-purpose building element composed of a 3.22 × 2.72 m Cross-Laminated Timber (CLT) frame and a glass infill.

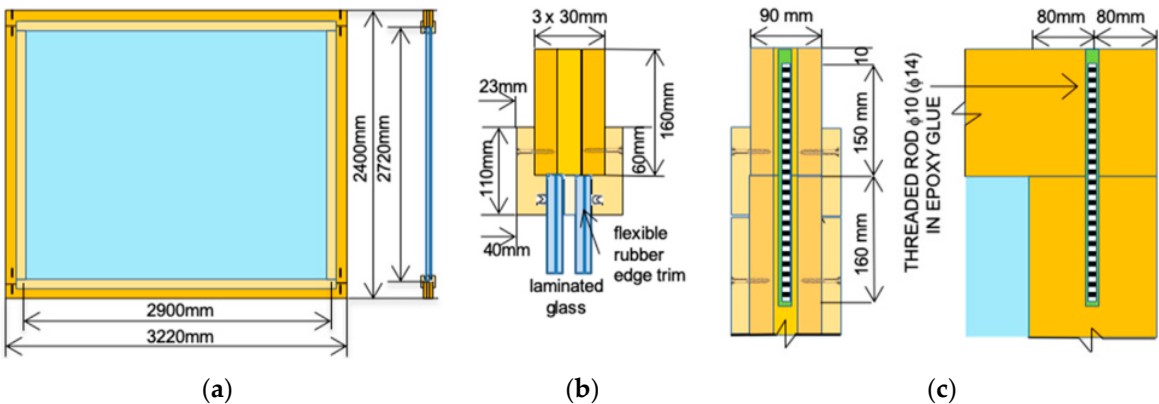

**Figure 2.** Full-size CLT-glass façade element: (**a**) front view, (**b**) CLT-to-glass frame connection detail (cross-section), and (**c**) timber frame corner joint. Figure reproduced from [9] with permission from Elsevier, Copyright© license number 4803680633411, April 2020.

The design concept for this hybrid system was developed through the optimal mechanical combination of two relatively new constructional materials, CLT and glass. On the one side, it is recognized worldwide that wooden buildings can benefit from the versatility of timber and its seismic, fire, thermal, and acoustic potential. For CLT, major applications take the form of floor and wall panels for high-rise buildings [10,11]. On the other hand, glass has been also increasingly used in buildings over the last decade. While glass products are well covered by European standards [12,13], harmonised European standards for structural designs are still in preparation [14–16], and some design issues can be solved with the support of technical guidelines (see [17–19]). One of the most challenging issues for load-bearing glass elements is their thermal performance assessment and potential effect on mechanical parameters [20–23].

The hybrid panel in Figure 2a was structurally optimized by including a series of b × h = 90 × 160 mm CLT framing members and two laminated glass plates (Figure 2b). A two-ply semi-tempered section was used for each of the glass panels and was obtained by bonding 10 mm sheets and a middle 1.6 mm thick EVASAFE® layer. A continuous flexible rubber edge trim was then used to seal the interposed air cavity (see Figure 2b) with s = 12.8mm the cavity thickness. The mechanical efficiency of the full-size façade system designed in [7–9] was found on friction mechanisms only, without the need

for adhesive or mechanical connections at the CLT-to-glass interface. Accordingly, the glass layers were designed with trapezoidal polished edges to minimize the possible micro-cracks and weak spots under operational conditions.

The extended full-size experiments reported in [7–9] provide evidence for the material's good load-bearing capacity for static and dynamic loads agreeing with [24], as well as its stability, serviceability, resistance, and excellent ability to dissipate input seismic energy. Moreover, it is well known that a given façade solution must fulfill a multitude of performance requirements, including thermal response, energy efficiency, waterproofness, airtightness, etc. Accordingly, the thermal performance characterization for the façade element in Figure 2 was considered in the framework of the VETROLIGNUM project (www.grad.unizg.hr/vetrolignum), which is also partly discussed in this paper.

## 2. Thermal and Energy Efficiency Assessment of Façade Components

The thermal strategic role of façades is a crucial step in research and design [25], especially for non-traditional envelopes that propose the use of new constructional details, for which various literature studies can be found in [26–29]. Even more extended calculation efforts may be required for the so-called "adaptive" dynamic systems that are subjected to continuous variations in their boundary conditions and performance [30]. Among other things, the use of timber in façades has been explored both in the form of secondary non-structural cladding elements [31] and as load-bearing frames.

### 2.1. Reference Thermal Performance Indicators

For the present study, the EN ISO 13788 standard was followed [32]. This standard is focused describing the assessment of the hygrothermal performance of building components and building elements. This standard recommends calculating the well-known temperature factor at the internal surface $f_{\text{Rsi}}$ as

$$f_{Rsi} = \frac{T_{si} - T_{out}}{T_{int} - T_{out}} \tag{1}$$

where $T_{si}$ is the temperature of the internal wall surface, $T_{int}$ is the internal air temperature, and $T_{out}$ is the external air temperature. $T_{si}$ strictly depends on the features of the investigated building system and can be especially sensitive to thermal bridges. As is known, $f_{\text{Rsi}}$ should be close to the unit to represent optimally insulated buildings. To ensure the occurrence of mould growth and surface condensation in dwellings, the $f_{\text{Rsi}} \geq 0.75$ values are commonly accepted in practice. In addition to the EN ISO 13788 recommendations, a series of National guidelines are available in several countries that recommend minimum limit values for $f_{\text{Rsi}}$. These are in the range of $f_{\text{Rsi}} \geq 0.52$ (France), $f_{\text{Rsi}} \geq 0.65$ (The Netherlands), or $f_{\text{Rsi}} \geq 0.7$ for Germany [33].

Another relevant parameter is the internal surface resistance, which depends on convection and radiation coefficients, air movements in the room, the air and surface temperature distribution in the room, and the surface material's properties. Accordingly, refined and time-consuming numerical models of a given room as a whole are essential to account for several aspects such as the thermal resistance of the surrounding envelopes, the environmental temperature, the air distribution in the room, and the room's geometry. However, simplified calculation methods or input values recommended by the existing guidelines can be used for preliminary estimates.

In an outdoor climate (temperature and relative humidity), four main parameters control the surface condensation and development of fungi [32]:

- The "thermal quality" of the peripheral elements of a building is represented by thermal resistance, thermal bridges, geometry, and internal surface resistance; these are defined by the temperature factor on the inner surface, $f_{\text{Rsi}}$;
- The internal humidity, influencing the dew point in the air;



- The indoor air temperature: A lower room temperature is generally more critical for rooms with reduced intermittent heating, or unheated rooms where water vapour can escape from adjacent heated rooms;
- And heating systems, that affect air movement and temperature distribution.

However, many problems can arise and should be properly assessed for novel envelope systems, such as:

(a) Mould that can occur when the surface humidity is higher than 80% over several days [33,34];

(b) Troubles due to danger of material corrosion: For each part of a heated building, it should be checked that the expected $f_{Rsi}$ value on the interior surface is less than the maximum allowable limit [35,36];

(c) A surface temperature that is too low, as resulting from contacting exterior surfaces of heated spaces, poor external thermal insulation of building sections, possible thermal bridges, or furniture arrangement [37,38];

(d) An increase in humidity, as resulting from an increase in the air humidity (at the air contact position with the wall) above the recommended design values (i.e., inappropriate use of spaces and airflows, see for example [39,40]).

### 2.2. Reference Ambient Conditions

As far as Equation (1) is considered for experimental studies, both the envelope features and the climate conditions for a given envelope/region can be responsible for the relevant design decisions and classification rules. Köppen's climate description for the Croatian region [41], for example, revealed a moderately warm rainy climate for the largest National area, with the mean monthly temperature in the coldest month of the year above −3 °C and below 18 °C (up to 22 °C in the hottest periods of the year).

For the VETROLIGNUM project, field ambient measurements have been monitored since early 2018 and have provided evidence of typical temperature records (see Figure 3), thus providing another parameter to be considered for thermal performance assessment.

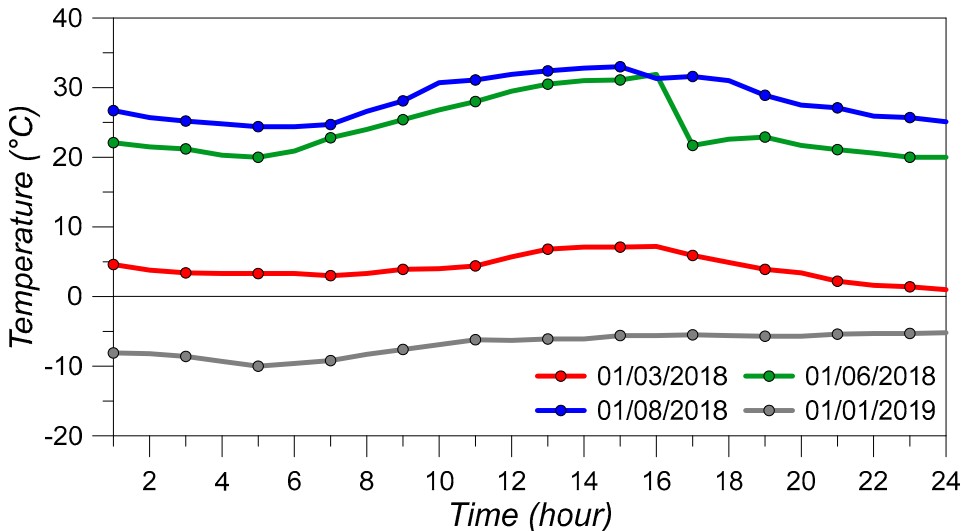

**Figure 3.** Example of meteorological data (selection of outdoor temperature acquisitions) from the Municipality of Zagreb (Faculty of Civil Engineering).

## 3. Mock-up Building

Given the design challenges and requirements summarized in Section 2, this paper experimentally and numerically explores the thermal performance and energy efficiency of a full-size building

prototype that is currently under investigation within the framework of the VETROLIGNUM project. A mock-up building prototype including hybrid CLT–glass façade elements (herein called the "Live-Lab" facility) was constructed at the beginning of 2018 and is still collecting relevant data.

The facility (Figure 4) consists of a full-sized single space characterized by plan dimensions B = 3.22 m × W = 2.80 m, with H = 2.8 m as the height. The 3D building is located on the roof of a single-story building (not accessible to the public) at the University of Zagreb, Faculty of Civil Engineering. The transversal envelopes of the facility (West and East sides) currently consist of CLT–glass modules, as seen in Figure 2. However, the prototype is assembled in such a way that it allows removal of the CLT–glass façade elements (for future re-use in different mock-up configurations) and the introduction of solid timber walls (Figure 4a).

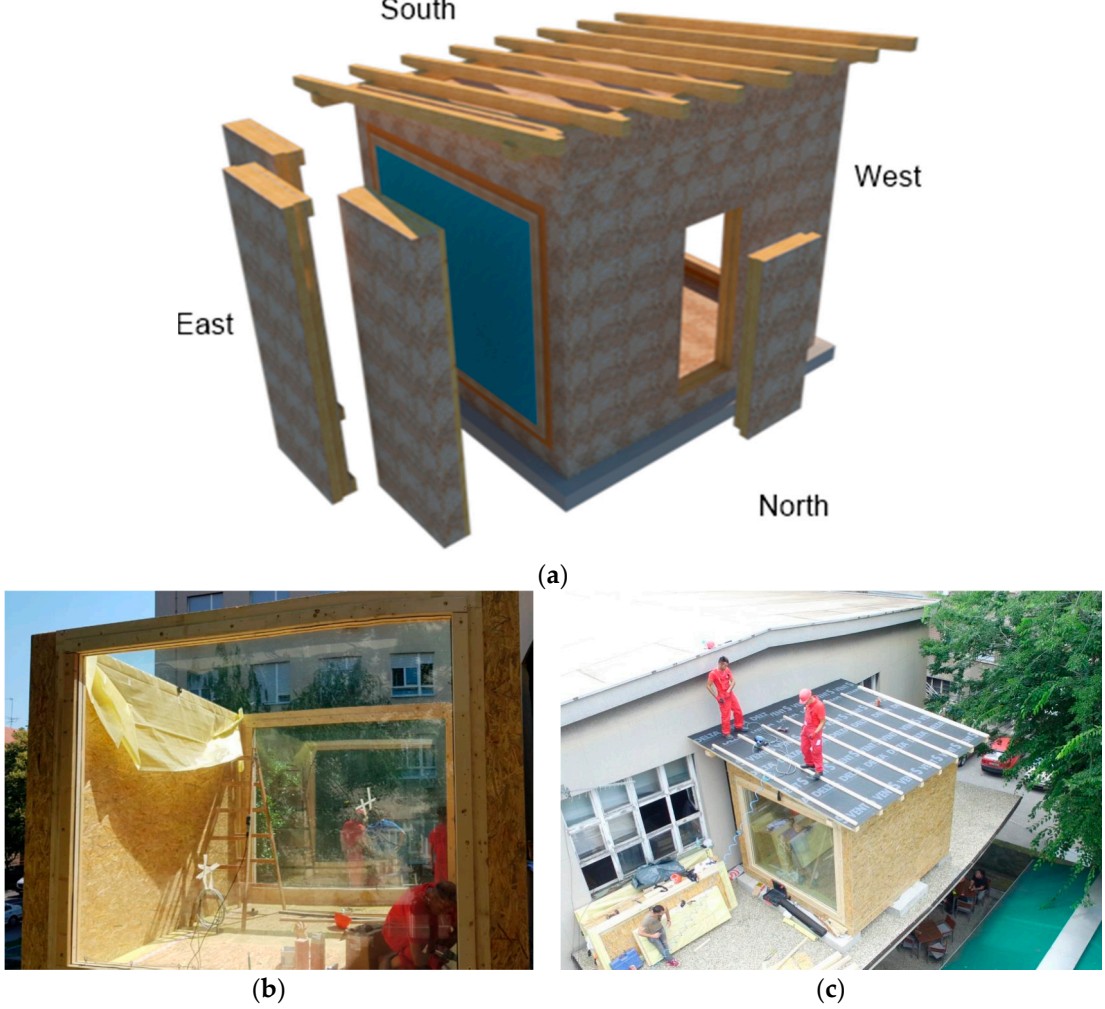

**Figure 4.** Assembly process for the Live-Lab facility at the University of Zagreb (2018): (**a**) the general concept; (**b**) CLT–glass façade elements and (**c**) roof installation.

The geometry and detailing of the main building components were defined on the basis of preliminary calculations for the expected load-bearing capacity under ordinary loads. This choice resulted in the use of prefabricated panels for the wooden houses to obtain a lightweight framework system. In Figure 5, a typical cross-section of the walls, the floor, and the roof panels is shown. The constituent layers include:

- 18 mm thick Oriented-Strand Board (OSB) panels (125 × 2500 mm their size) on both the internal and external sides;

- 200 mm wide timber frame members (KVH solid structural timber members marked by C24 resistance class (spruce) according to EN 338 provisions [42] dried to up to 18% humidity);
- A mineral wool insulation infill (NaturBoard VENTI mineral wool boards (600 × 1000 mm) by KNAUF Insulation [43]).

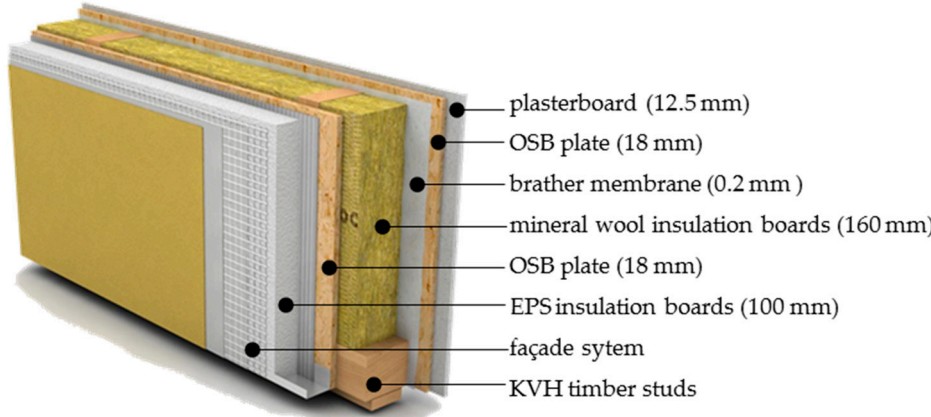

**Figure 5.** Typical layers for the prefabricated panels used in the mock-up facility.

A steam dam was also installed between the inner OSB and timber frame members (Delta-Dawi GP [44]). Finally, in contrast to Figure 5 and the common practice for buildings, additional polystyrene insulation layers (on the exterior side) or plasterboard foils (on the inner side) were omitted due to their lack of aesthetic requirements. Accordingly, an impregnating finisher was used on the OSB surfaces to ensure their water resistance.

The South longitudinal wall was erected with a single full-size panel. On the opposite side, the North wall was composed of two partitions to account for a possible door (Figure 4a). Two adjacent panels, mechanically interconnected, were used to realize the floor. For the single-pitch roof, timber beams were used to support the ceiling panels. Globally, the 3D assembly was thus characterized by the geometrical surface parameters reported in Tables 1 and 2.

**Table 1.** Building elements of the zone on the boundary heated outside.

| Exterior Walls of the Heated Section with Openings (m$^2$) | | | |
|---|---|---|---|
| North | East | South | West |
| 10.98 | 9.82 | 10.98 | 9.82 |
| **Heated Exterior Walls without Openings (m$^2$)** | | | |
| North | East | South | West |
| 10.98 | 0.00 | 9.38 | 0.00 |

**Table 2.** Openings (number and size) of the facility (cardinal directions).

| | | Heated | | | Outside | |
|---|---|---|---|---|---|---|
| Orientation | Opening Mark | N.° of Elements | Width (m) | Height (m) | Surface (m) | Total (m$^2$) |
| East | Façade | 1 | 3.61 | 2.72 | 9.8192 | 9.8192 |
| North | Door | 1 | 0.8 | 2 | 1.6 | 1.6 |
| West | Façade | 1 | 3.61 | 2.72 | 9.8192 | 9.8192 |
| | | | | | Total area of openings | 21.24 |

## 4. Testing Program

The experimental measurements for the 3D building prototype in Figure 4 were derived from the collection of one-year cyclic records, hereafter referred to as "Cycle 1" (September 2018–August 2019), and so on. These included:

- Indoor Relative Humidity (RH) and temperature;
- Outdoor RH and temperature;
- RH and temperature data within the cavity of the double insulated glass (for limited time intervals only);
- And preliminary measurements of the energy consumption for the 3D building system.

Alongside the ambient measurements, a thermographic camera was used to capture possible critical details of the Live-Lab assembly, paying special attention to the area of the corner joints.

### 4.1. Instruments

The external climate parameters (wind speed, wind direction, precipitation, temperature, and humidity) in the Live-Lab context were measured by means of a meteorological station (Figure 6). This was combined with a central control station designed by SITEL Ltd. (Ljubljana, Slovenia, www.sitel.si) to support the continuous monitoring of climate conditions and the acquisition of relevant data. As a specialty of the Live-Lab facility, a heating source was also considered for the cavity between the two glass panels. As the hygrothermal properties in the cavity were set as a control parameter for continuous measurement, the heater was set to automatically switch on at the first sign of the dew point for glass.

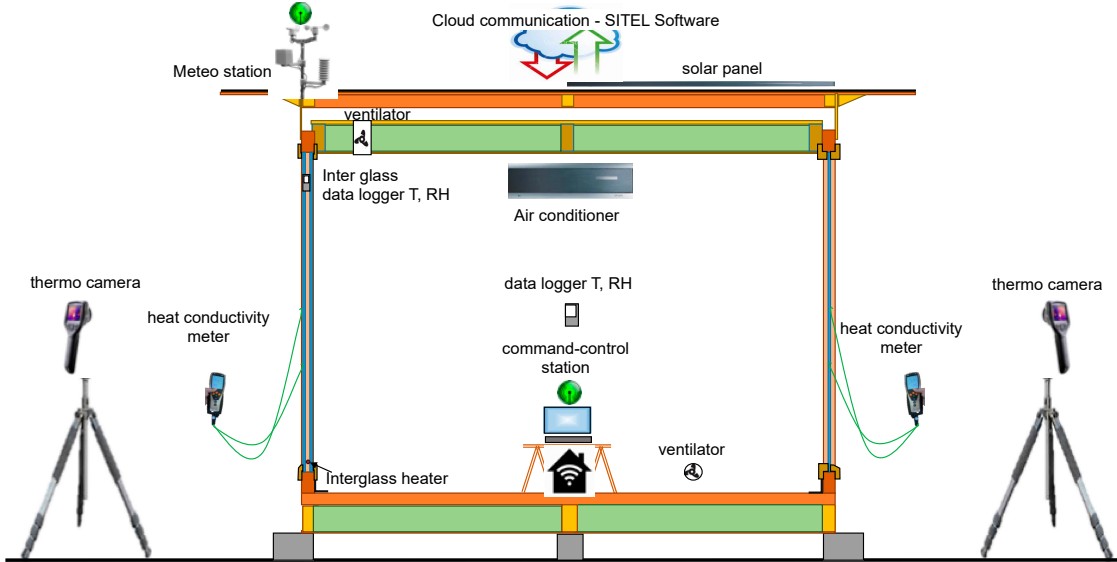

**Figure 6.** Instruments for the field experimental measurements (schematic cross-section of the Live-Lab mock-up building). RH, relative humidity.

In the post-processing stage, based on the collected measurement data, the impact of this kind of heating source was assessed in terms of the sensitivity of the thermal transmission of the 3D facility as a whole. This was also assessed in terms of its impact on the energy consumption to maintain a constant indoor temperature for the facility. To optimize the energy consumption of the Live-Lab prototype, solar panels were installed on the roof to act as an additional energy source. Electricity for heating or cooling the facility was thus supplied from both the electricity grid and from the solar panels. The setup instruments included an air conditioner and a fan to reproduce a reliable configuration for a residential building.

### 4.2. Experimental Records and Aquisitions

In Figure 7, an example of typical outcomes from the continuous monitoring of the Live-Lab facility are proposed in terms of indoor temperature and RH records (data acquisition from the SITEL control station). Depending on the variable external conditions (Figure 3), major efforts were made to achieve comfortable hygrothermal levels inside the mock-up building in line with conventional residential building performance indicators. Sometimes, however, temporary interruptions of the devices and instruments in use gave evidence of locally scattered data, as shown for the selected data in Figure 7.

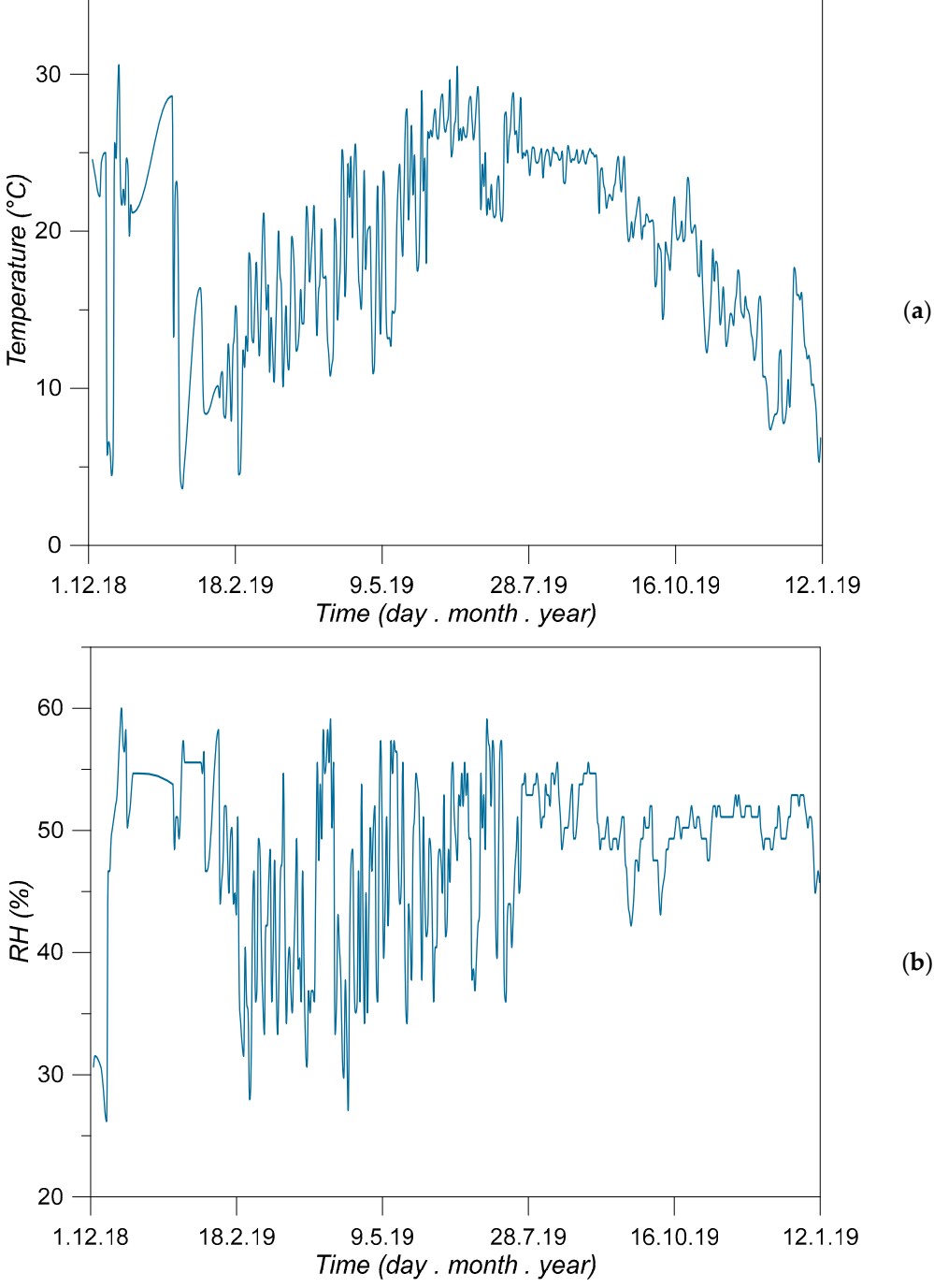

**Figure 7.** Live-Lab indoor measurements from the SITEL control station: examples of (**a**) temperature and (**b**) relative humidity.

Alongside the continuous record acquisitions, the thermographic images also provided evidence of linear thermal bridges (Figure 8) that were generally detected (as expected) in the transition regions from the CLT to glass members.

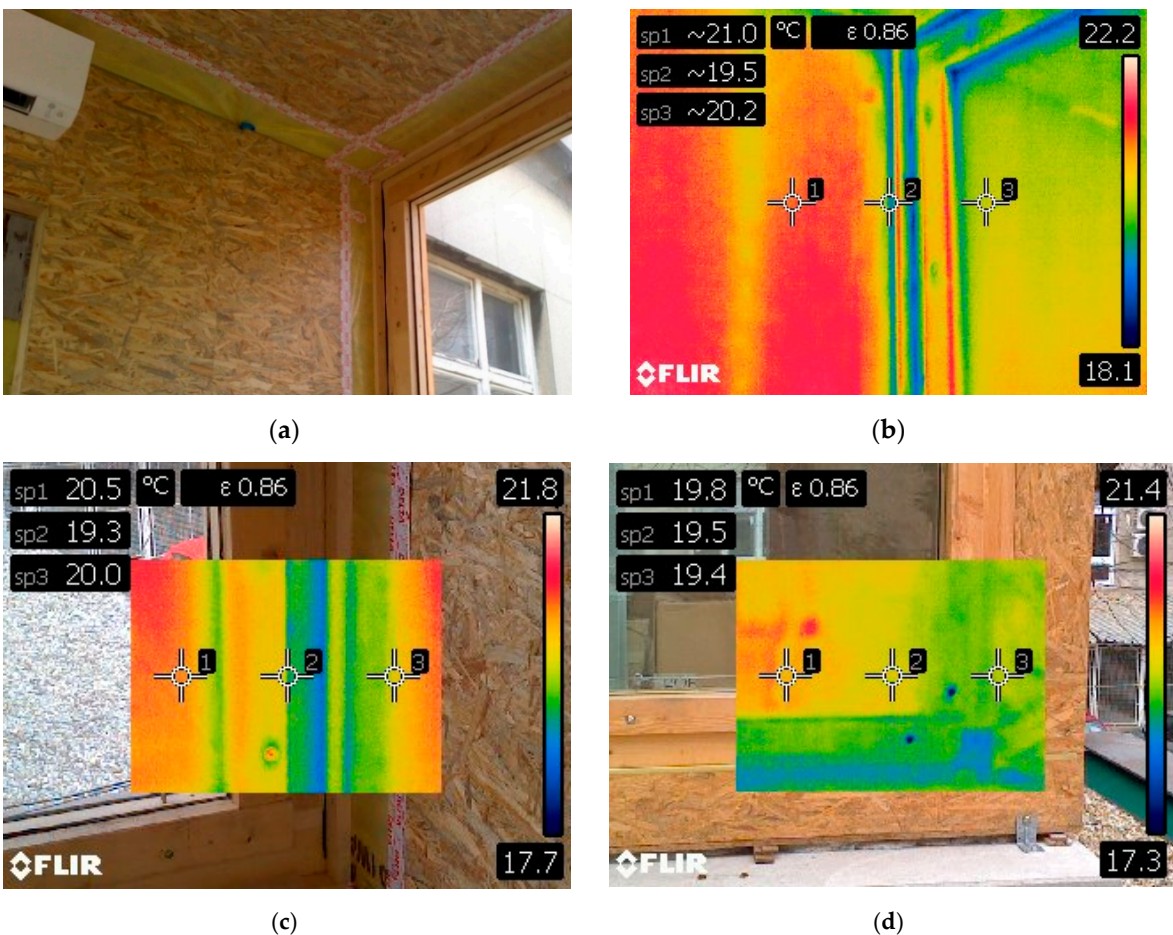

**Figure 8.** Thermal bridges detected by thermal camera acquisitions: (**a**) top corner layout (indoor) with the (**b**) corresponding temperature field and (**c**,**d**) bottom corner detail (indoor and outdoor measurements).

## 5. Full-Size Numerical Analysis of the Live-Lab Facility

### 5.1. Solving Approach and Input Data

Numerical simulations are presently one of the most powerful tools for design optimization and assessment. They are based on the concept that developing a virtual model (based on well-known physical processes and constitutive laws) consumes less money/time than developing a real experimental prototype. This applies especially to full-size structures and complex systems, such as the Live-Lab 3D facility investigated in this paper.

In the first stage of this study, a series of numerical simulations were carried out with the EnCert-HR computer software (www.encert.hr) to calculate—as accurately as possible—the expected overall energy consumption for the 3D Live-Lab facility in Figure 4. In doing so, the reference modelling procedure in Figure 9 was applied in the EnCert-HR software and adapted to the goals of this study.

For the Live-Lab facility, the numerical building prototype was assumed to be located at the Faculty of Civil Engineering of the University of Zagreb. Accordingly, the whole calculation process was developed with respect to the Croatian Technical Regulations on energy economy and heat retention in buildings (Official Gazette No. 128/15 [45]). The building characterization was also based on [46–48] and treated as an independent building unit. Special care was taken for the Northern side of

the Live-Lab, whose orientation toward the Faculty building was considered in the calculation by solar gains and losses. The calculation process also included the coefficient of heat loss due to ventilation.

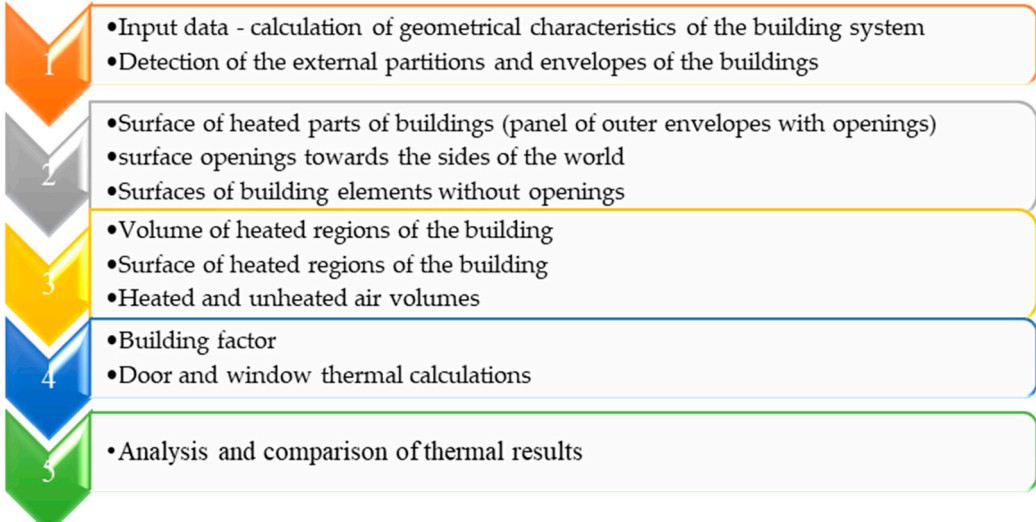

**Figure 9.** Schematic process for the thermal and energy analysis of the full-size facility (EnCert-HR).

As the basic step in the calculation process, the building components of the Live-Lab facility were first separately described and characterized (see Figure 10). These characterizations included two full-size CLT–glass façade elements (East and West sides), two solid walls according to Figures 4 and 5, and the floor and roof panels. A door opening was also placed on the North side.

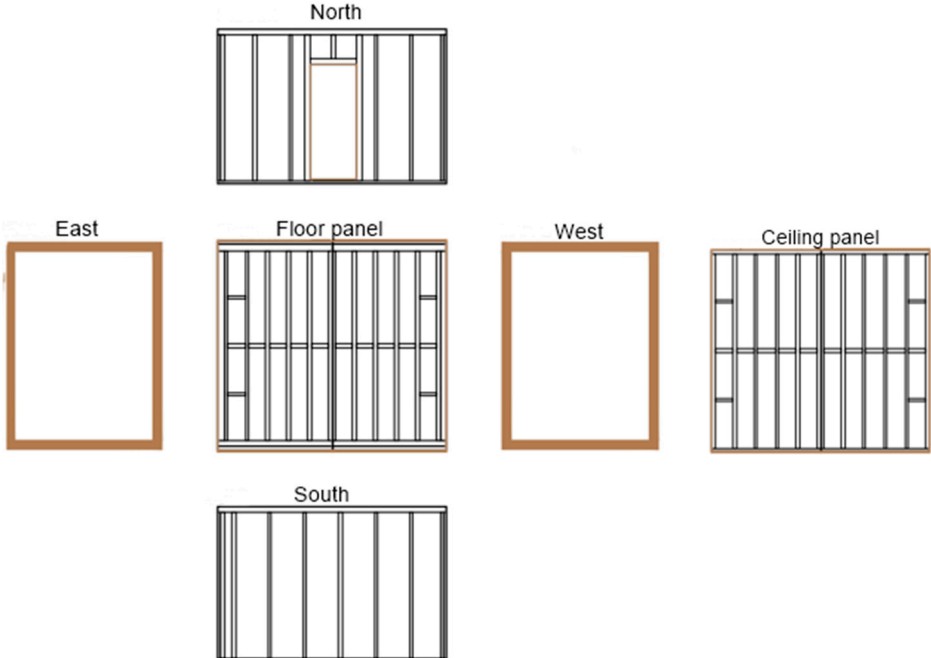

**Figure 10.** Arrangement of the building envelope components for the Live-Lab facility (EnCert-HR).

Through the calculation steps, the nominal geometrical and thermo–physical properties were determined for the materials and building components used (see Section 3).

The final advantage was then derived from the available Live-Lab meteorological data. These data included ambient and consumption records (for the energy efficiency assessment) and outdoor

average records (for the definition of realistic configurations in the definition of thermal performance indicators) (see Table 3).

**Table 3.** Reference meteorological data for Zagreb city (average Live-Lab outdoor records).

| | Month | | | | | | | | | | | |
|---|---|---|---|---|---|---|---|---|---|---|---|---|
| | **Jan** | **Feb** | **Mar** | **Apr** | **May** | **Jun** | **Jul** | **Aug** | **Sep** | **Oct** | **Nov** | **Dec** |
| Temperature (°C) | 1.0 | 2.9 | 7.1 | 11.7 | 16.8 | 20.3 | 21.9 | 21.3 | 16.3 | 11.4 | 6.5 | 1.4 |
| Relative humidity (%) | 81.0 | 74.0 | 68.0 | 67.0 | 66.0 | 67.0 | 67.0 | 69.0 | 76.0 | 80.0 | 83.0 | 85.0 |

*5.2. Preliminary Energy Efficiency Assessment*

First, besides the simplified geometrical description of the envelope details for the full-sized numerical model of the Live-Lab prototype, the required energy for heating and cooling the facility was calculated (see Figure 11).

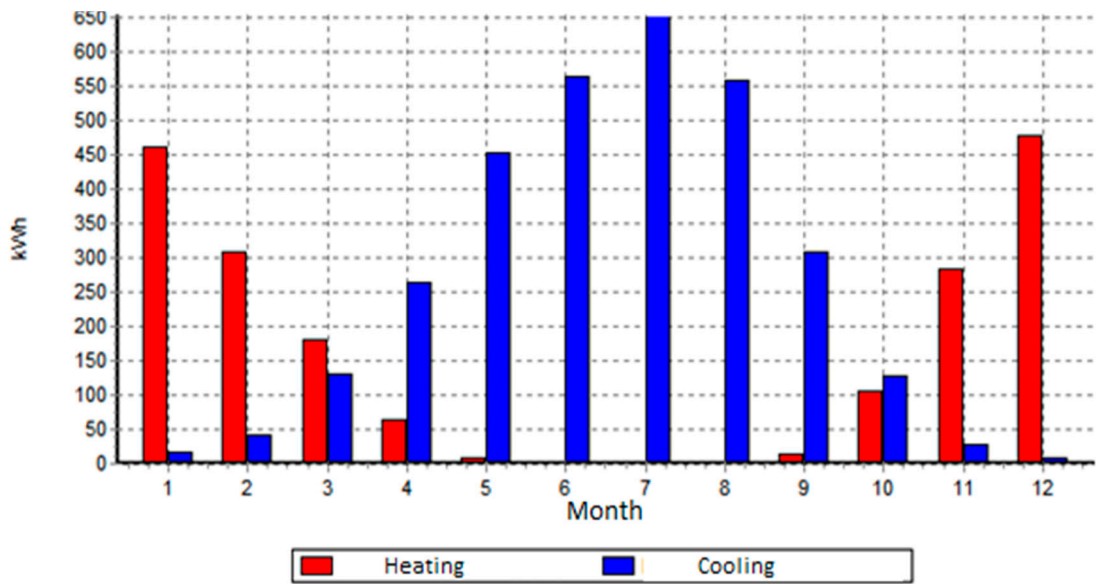

**Figure 11.** Calculation of required heating and cooling energy (EnCert-HR).

The expected cooling energy was found to reach the highest values from April to September, with $Q_C$ = 260–650 kWh. Similarly, the maximum energy required for heating was expected from November to February, with $Q_H$ = 300–460 kWh. The annual energy for heating and cooling was thus estimated as $Q_{H,nd}$ = 1906 kWh/year and $Q_{c,nd}$ = 3155 kWh/year, respectively, with a relatively higher cooling consumption. Significant energy losses, as expected, were observed to derive especially from the linear thermal bridges due to the different properties of the materials used, as well as from relevant geometrical variations (Figure 8).

The energy estimates in Figure 11 refer to the floor area of the building prototype. The maximum requirements per year/square meter were found to be ≈128 kWh/m$^2$ year and ≈210 kWh/m$^2$ year for heating and cooling, respectively.

The above estimates are related both to the geometrical aspects of the facility and to climatic conditions. Following Figure 10 and the data previously reported in Tables 1 and 2, the shape factor $f_0$ of the building was also calculated using the most common index in professional practice. The latter is commonly defined as the ratio between the envelope surface ($A$) and the inner heated volume ($V_c$) of a given building, which for the Live-Lab mock-up facility is

$$f_0 = \frac{A}{V_e} = \frac{71.4m^2}{46.49m^3} = 1.54\ m^{-1}. \tag{2}$$

The research in [49,50] notes that the above shape factor can affect the energy demands of a given building, especially one with wide glass surfaces and complex architectural geometries and irregular façades. The parametric study described in [49] for five building examples (Nordic climate conditions) with imposed $f_0$ values in the range from 1 to 1.7, for example, resulted in up to +10%–20% energy demand variations for higher shape factors. Accordingly, this aspect should be further assessed in future developments of the Live-Lab facility. Given the value in Equation (2), additional relevant geometrical characteristics are represented by

- Net volume $V = 31.33$ m$^3$;
- Gross floor area $A_{floor} = 14.95$ m$^2$.

Thus resulting in a glass-to-floor-area ratio of $\approx 1.14$.

The examined 3D prototype is characterized by a compact regular shape/size that cannot represent a real building configuration but can still efficiently support research aimed at the optimization of the hybrid CLT–glass façade concept, working towards the definition of generalized design recommendations. A notable intrinsic feature of the CLT–glass design concept is the relatively large glass-to-floor-area ratio (with $\approx 0.2$–0.25 as the conventional value; see [49]) for the façade elements directly exposed to external environmental events.

### 5.3. Thermal Characterization and Performance Assessment

The simplified EnCert-HR numerical model was further investigated to assess the thermal performance indicators for the same Live-Lab.

### 5.3.1. Thermal Characterization of the Building Sections (Prefabricated Panels)

The heat transfer coefficient was calculated for the prefabricated panels in use to verify that the requirement for dynamic thermal performance could be met with respect to the reference limit conditions ($U_{max} = 0.30$ W/m$^2$K). Following the schematic cross-section in Figure 5, the thermo–physical properties in Table 4 were used for calculations.

**Table 4.** Nominal layout and thermo–physical properties for the prefabricated panels [51].

| Material | Thickness | Specific Heat Capacity | Density | Thermal Conductivity | Differential Resistance |
|---|---|---|---|---|---|
| | $d$ (cm) | $c_p$ (J/kgK) | $\rho$ (kg/m$^3$) | $\lambda$ (W/mK) | $S_d$ (m) |
| 4.09—directional chipboards (OSB) | 1.80 | 1700 | 650 | 0.130 | 0.9 |
| DELTA-DAWI-GP—vapour barrier | 0.02 | 1250 | 180 | 0.190 | 100 |
| 7.01—mineral wool (MW) | 20.00 | 1030 | 30 | 0.040 | 0.2 |
| 4.09—directional chipboards (OSB) | 1.80 | 1700 | 650 | 0.130 | 0.9 |
| Total | 23.62 | | | | 102 |

The heat transfer coefficient $U$ for the building part was obtained based on:

- The surface resistance, with $R_{si} = 0.13$ m$^2$K/W and $R_{se} = 0.04$ m$^2$K/W for the internal and external sides, respectively (see also Table 5);
- The thermal resistance of the constituent homogeneous layers, as summarized in Table 4:

$$R_T = R_{si} + \sum \frac{d_i}{\lambda_i} + R_{se} = 5.45 \frac{m^2 K}{W}, \tag{3}$$

resulting in:

$$U = \frac{1}{R_T + R_u} + \Delta U = 0.18 \frac{W}{m^2 K} < U_{max} = 0.30 \frac{W}{m^2 K} \tag{4}$$

where $R_u = 0$ is the thermal resistance for attic spaces.

**Table 5.** Surface condensation risk assessment for the building part.

| Month | Vapour Pressure in Space | Saturated Vapour Pressure | Internal Air Temperature | External Air Temperature | Surface Temperature | Temperature Factor (Equation (1)) |
|---|---|---|---|---|---|---|
| | $p_i$ (kPa) | $p_{sat}$ (kPa) | $T_{int}$ (°C) | $T_{out}$ (°C) | $T_{si,min}$ (°C) | $f_{Rsi}$ |
| Jan | 1.075 | 1.344 | 20.0 | 1.0 | 11.4 | 0.547 |
| Feb | 1.119 | 1.399 | 20.0 | 2.9 | 12.0 | 0.532 |
| Mar | 1.218 | 1.522 | 20.0 | 7.1 | 13.3 | 0.480 |
| Apr | 1.396 | 1.745 | 20.0 | 11.7 | 15.4 | 0.446 |
| May | 1.778 | 2.222 | 20.0 | 16.8 | 19.2 | 0.750 |
| Jun | 2.058 | 2.572 | 20.0 | 20.3 | 21.6 | - |
| Jul | 2.058 | 2.572 | 20.0 | 21.9 | 21.6 | - |
| Aug | 2.058 | 2.572 | 20.0 | 21.3 | 21.6 | - |
| Sep | 1.737 | 2.171 | 20.0 | 16.3 | 18.8 | 0.675 |
| Oct | 1.376 | 1.720 | 20.0 | 11.4 | 15.1 | 0.430 |
| Nov | 1.204 | 1.504 | 20.0 | 6.5 | 13.1 | 0.489 |
| Dec | 1.084 | 1.355 | 20.0 | 1.4 | 11.5 | 0.543 |

Once the capacity of the mineral wool prefabricated panels was verified (using Equation (4)) to meet the minimum requirements, the design temperature factor was set as [52]

$$f_{Rsi,design} = \frac{R_T - R_{si}}{R_T} = 0.982 \tag{5}$$

and used to assess potential condensation risk.

The major results are summarized in Table 5 and Figure 12, where it is possible to see that the expected diffusion flow of water vapour through the walls is rather stationary [53] for the examined climatic conditions. Furthermore, Table 6 shows that the calculated $f_{Rsi}$ values are in close agreement with the internationally recommended values reported in Section 2. The design temperature factor calculated in Equation (5) is also greater than the maximum required value for $f_{Rsi}$, further enforcing the good thermal performance estimates.

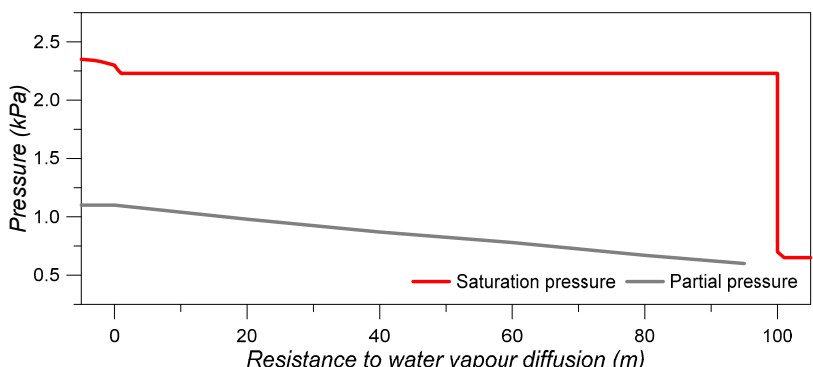

**Figure 12.** Internal condensation—water vapour pressure in the building part.

### 5.3.2. Thermal Characterization of the Façade and Door Openings

Finally, the EnCert-HR model was used to determine the thermal performance of the hybrid façade element object of analysis. Based on the Technical Regulations in use, the overall impact of linear thermal bridges was also considered by increasing the predicted $U$ value by an $U_{TM} = 0.05$ W/m²K increment. The derived thermal indicators are presented in Table 6.

**Table 6.** Calculation of the thermo–physical parameters for the façade and door openings.

| Parameter | Symbol | Unit | Façade Element | Door |
|---|---|---|---|---|
| Total heat transfer coefficient | $U$ | (W/m²K) | 2.42 | 0.43 |
| Degree of transmission of solar energy through glazing | $g_\perp$ | - | 0.64 | 0 |
| Opening surface | $A_w$ | (m²) | 9.82 | 1.78 |
| Proportion of glazing | $1\text{-}f_f$ | - | 0.8 | 0.7 |
| Panel | $f_p$ | - | 0 | 0 |
| Frame surface | $A_f$ | (m²) | 1.964 | 0.534 |
| Glazing surface | $A_g$ | (m²) | 7.856 | 1.246 |
| Panel surface | $A_p$ | (m²) | 0 | 0 |
| External glazing heat transfer coefficient | $U_{g1}$ | W/m²K | 5.15 | 1 |
| Degree of transmission of solar energy through glazing | $g_{\perp 1}$ | - | 0.80 | 0.00 |
| Internal glazing heat transfer coefficient | $U_{g2}$ | (W/m²K) | 5.15 | - |
| Degree of transmission of solar energy through glazing | $g_{\perp 2}$ | - | 0.80 | - |
| Heat transfer coefficient of the frame | $U_f$ | (W/m²K) | 1.8 | - |
| Heat transfer coefficient of glass | $U_g$ | (W/m²K) | 2.51 | 0.18 |
| Heat transfer coefficient of the panel | $U_p$ | (W/m²K) | 0 | 0 |
| Glazing perimeter | $l_g$ | (m) | 11.54 | 0 |
| Linear thermal bridge (glass edge) | $\psi_g$ | (W/mK) | 0.05 | 0.05 |
| Panel perimeter | $l_p$ | (m) | 0 | 0 |
| Linear thermal bridge (panel edge) | $\psi_p$ | (W/mK) | 0 | 0 |
| Opening inclination (to horizontal) | $\alpha$ | (deg) | 90 | 90 |
| Internal surface resistance | $R_{si}$ | (m²K/W) | 0.13 | - |
| External surface resistance | $R_{se}$ | (m²K/W) | 0.04 | - |
| Air cavity resistance | $R_s$ | (m²K/W) | 0.18 | - |

For the *U*-value for CLT–glass façade elements in Table 6, a relatively high parameter is found compared to glass façade systems in general, with a recommended *U*-vale in the range of 1.6 W/m²K (see also [28]). This latter estimate, however, could be justified by the lack of geometrical details in the full-size EnCert-HR model, thus requiring further local investigations.

## 6. Thermal Numerical Analyses of the Hybrid CLT–glass Component Level

Besides the promising data summarized in Section 5, a further study was successively carried out in the form of a more accurate thermal numerical analysis of the typical CLT–glass façade element in use at the Live-Lab facility of Figure 4. As shown in Section 5, commercial software codes that are suitable for design purposes can be computationally efficient for full-size calculations in professional applications. However, at the same time, they have intrinsic limits in the geometrical detailing of building components and on the reliability of local estimates. This is also the case for the examined hybrid CLT–glass façade elements, with expected local criticalities, such as the corner joints, depicted in Figure 8.

### 6.1. Methods and Assumptions

A refined numerical study was dedicated to the thermal analysis of the façade elements only. Taking advantage of the double symmetry of the reference CLT–glass panel in Figures 2 and 4, the corresponding Finite Element (FE) model was described in ABAQUS [54], which primarily focused on $\frac{1}{4}$th the nominal module (with appropriate boundary conditions). The final FE assembly (see Figure 13) consisted of a set of 8-node heat transfer solid elements (DC3D8 type from ABAQUS library) comprising:

- Two double laminated glass sections (10 + 10 mm glass layers with a middle 1.6 mm thick EVASAFE bond);
- The interposed air cavity (*s*= 12.8 mm the thickness);
- The linear rubber edge trim allowing the glass panels to keep their position under ordinary thermo–mechanical loads,
- The CLT frame members (90 × 160 mm their cross-section);

- Additional timber purlins, providing a linear slot for the positioning of the glass panels and thus being responsible for the activation of the frictional mechanisms along the CLT–glass edges in contact.

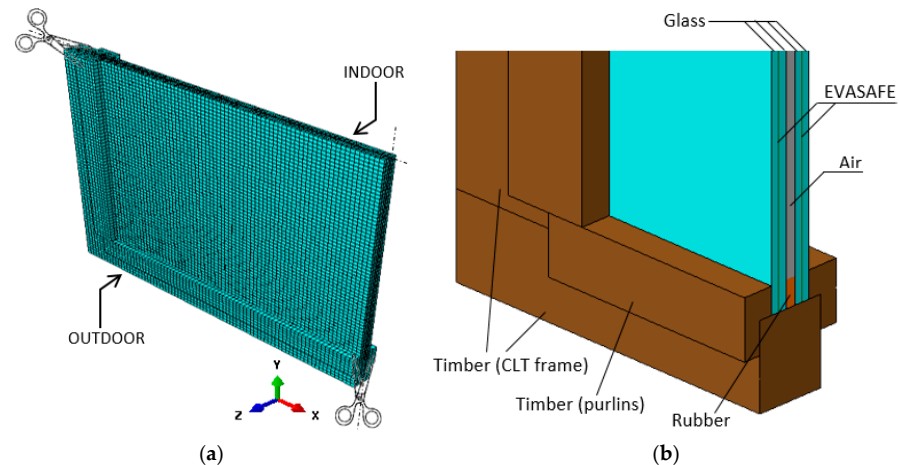

**Figure 13.** Reference FE model for the thermal assessment of full-size CLT–glass façade elements: (**a**) global assembly (1/4th the geometry) and (**b**) a detailed view with hidden mesh (ABAQUS).

Similar to Section 5, the thermal performance of the CLT–glass façade panel was first assessed by evaluating a few key performance indicators, such as the overall heat transfer coefficient $U$ and the corresponding linear thermal transmittance (Ψ-value). Further advantages of the detailed FE model were then represented according to the availability of a series of local estimations of temperature values that are primarily involved in the thermal characterization of façade systems, such as the expected temperature under an imposed thermal gradient $\Delta T$ (for condensation risk assessment) and the minimum surface temperature on glass surfaces, as well as the relevant temperature data that are directly involved in the definition of the overall thermal comfort (i.e., $f_{Rsi}$ from Equation (1)). In doing so, the characterization of materials was carried out based on the nominal thermo–physical parameters provided by the design standards and literature references (Table 7).

**Table 7.** Input thermal properties in use for FE modelling in ABAQUS.

| | **Component** | | | **Interposed Cavity** | |
| --- | --- | --- | --- | --- | --- |
| | **Glass** | **EVASAFE** | **Timber** | **Rubber** | **Air** |
| Conductivity λ (W/mK) | 0.8 | 0.19 | 0.3 | 0.1 | 0.028 |
| Emissivity ε (−) | 0.95 | / | 0.7 | / | / |

Special care was paid to the air volume enclosed in the cavity of laminated glass panels, which was properly described as the equivalent conductivity, including the effects of the imposed temperature gradient ($\Delta T$) and the corresponding gas properties (see also [28]).

Finally, the thermal boundary conditions for the FE steady-state simulations were defined with the support of a set of "surface film" and "surface radiation" thermal interactions from the ABAQUS library—both for the internal and external surfaces of the glass panels and the timber members.

### 6.2. Expected Thermal Performance Indicators

Based on EN ISO 10077-2:2017 [55] provisions, the thermal calculation for the system in Figure 13 should be conventionally carried out based on:

- Relative Humidity= 50%,
- External condition: $T_{out}$= 0 °C, film coefficient (timber and glass) = 23 W/m$^2$K,

- Internal condition: $T_{int}$= 20 °C, film coefficient (timber and glass) = 8.02 W/m$^2$K.

This recommended thermal scenario was found to be representative of real field measurements from the Live-Lab facility (see Section 3). The major FE outcomes from the above thermal boundaries are presented in Table 8, where a significant improvement in the expected *U*-value from Section 5 can be found (with $U$ = 2.42 W/m$^2$K in Table 6).

**Table 8.** Numerical thermal performance assessment of the CLT–glass façade element under an imposed $\Delta T$ = 20 °C (detailed FE model, ABAQUS). $T_{max}$ and $T_{min}$ values are calculated in the cavity.

| Parameter | Double Laminated Glass with EVASAFE |
|---|---|
| $U$ (W/m$^2$K) | 1.826 |
| $\Psi$ (W/mK) | 0.5285 |
| $T_{max}$ (°C) | 16.60 |
| $T_{min}$ (°C) | 9.15 |

If the nominal geometry of the load-bearing components is described with accuracy, the local estimates of temperature distributions can further support the thermal performance assessment (and possible optimization) of the reference CLT–glass system in Figure 13. Figure 14, in this regard, shows a typical temperature distribution in glass and the adjacent portion of frame (corner joint).

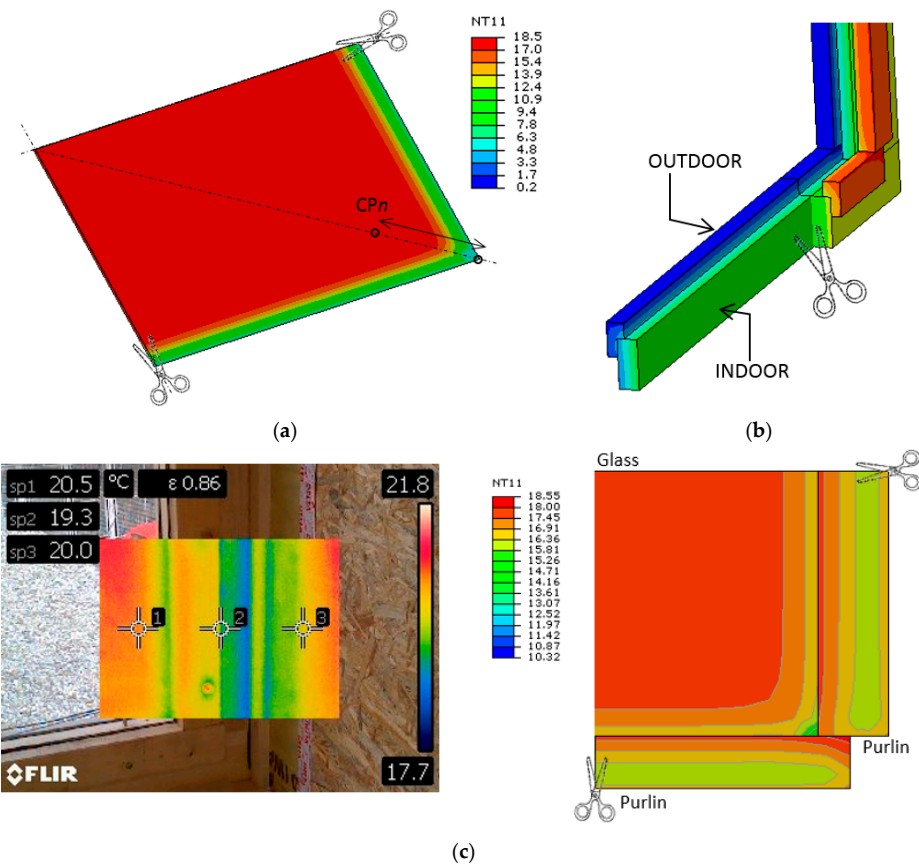

(a)

(b)

(c)

**Figure 14.** Numerical thermal performance assessment of a full-size CLT–glass façade element (ABAQUS). Expected distribution of temperature (**a**) in the internal glass panel (external side with CP*n* = 1, 2, and 3 diagonal distances corresponding to 50 mm, 90 mm, and 150 mm, respectively from the corner) and (**b**) in the timber frame with (**c**) a detail of the corner joint (front view). All the legend values are given in °C.

According to Figure 14a,b, the temperature rapidly changes in the area of glass in contact with timber, thereby affecting the key performance indicators. Figure 14c, moreover, demonstrates the qualitative correlation of this detailed FE estimation with the thermographic acquisitions from the Live-Lab facility.

Even more attention, however, should be paid to the temperature distribution in the frame cross-section (CLT members and purlins) given that the hybrid CLT–glass design concept is based on direct frictional contact between the two materials. The latter is emphasized in Figure 15a (timber members), while Figure 15b offers a more detailed representation of temperatures according to glass thickness and different locations.

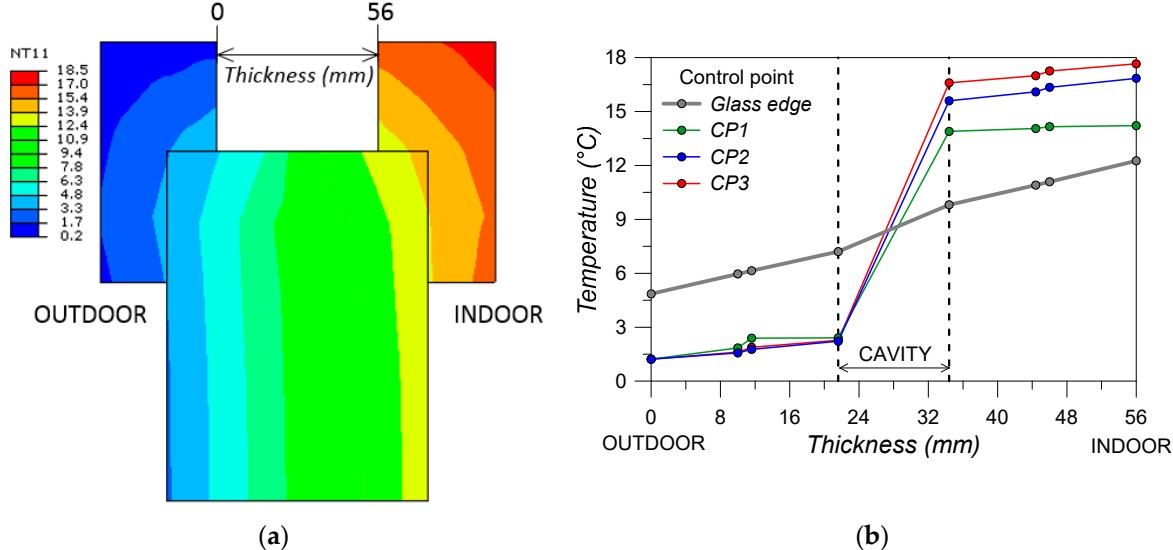

**(a)**　　　　　　　　　　　　　　　　**(b)**

**Figure 15.** Numerical thermal performance assessment of a full-size CLT–glass façade element. Temperature distribution: (**a**) in the frame cross-section (legend values in °C) and (**b**) in glass thickness (with CP*n* = 1, 2, and 3 diagonal control points at a distance of 50 mm, 90 mm, and 150 mm from the corner; ABAQUS).

As long as the glass components are in contact with the frame in Figure 15a, the temperature distribution at the glass edges corresponds to the "Glass edge" plot in Figure 15b. In the same figure, moreover, the CP1, CP2, and CP3 control points are still representative of temperature distributions in the thickness of the glass panel. According to the contour plot in Figure 14a, however, these CP*n* values are selected on the diagonal of the panel at a distance of 50 mm, 90 mm, and 150 mm from the corner.

To avoid condensation, the minimum surface temperature ($T_{min}$) on the internal face of glass (i.e., inside the cavity volume) needs to be higher than the dew point temperature ($T_{dp}$) corresponding to the air conditioning inside the panel:

$$T_{min} \geq T_{dp} = f(RH, T). \tag{6}$$

The minimum measured surface temperature for the FE configuration was calculated to be a minimum of 9.15 °C (corner edge in the frame area, see Table 8). According to the reference European standard, the $T_{dp}$ value should be checked based on mock-up experiments including the continuous monitoring of temperature and humidity values inside the cavity (see, for example, [56]). A similar approach was proposed in [26] for a novel curtain wall system with a metal frame, but this approach is a key design step for glass envelopes in general. Conservatively, the calculation step was first carried out in this paper for the worst condition in the cavity volume, corresponding to an air temperature of 9 °C and an RH = 80%. This assumption results in $T_{dp}$ = 5.7 °C and consequently suggests that the examined CLT–glass façade element has appropriate thermal performance (Equation (6)). Its overall

thermal behavior, however, will be further explored in the next stages of the research project with the support of extended mock-up registrations.

Another relevant parameter is then represented by the minimum surface temperature on the indoor face of the glass ($T_{si}$, with $T_{si}$ = 11.6 °C at the corner edge and $T_{si}$ = 14 °C at the exposed corner of the glass, see Figure 15b). Here, for indoor conditions of 20 °C (air temperature) and RH = 50%, the recommended reference value is $T_{dp}$ = 6 °C, thus suggesting again optimal satisfaction of the limit conditions for the CLT–glass solution.

### 6.3. FE Temperature Factor for the Hybrid CLT–glass Façade Element

The temperature factor was ultimately calculated at the component level based on Equation (1) and the FE numerical predictions for the system of Figure 13, as:

$$f_{Rsi,FE} = \frac{T_{si,FE} - T_{out,FE}}{T_{int,FE} - T_{out,FE}} = 0.613. \tag{7}$$

This provides a promising value compared to the minimum requirements reported in Section 2 and further enforces the potential of the hybrid CLT–glass solution. Moreover, Equation (7) still represents the average value calculated from an ideal $\Delta T$ = 20 °C thermal scenario (see Section 6.2).

Accordingly, even more accurate calculations for $f_{Rsi}$ were carried out (still at the component level only) by taking advantage of the meteorological records from the Live-Lab mock-up facility, assuming the average outdoor temperatures in Tables 3 and 5 as the reference input for a set of additional FE numerical calculations.

In doing so, the FE assembly of Figure 13 was thus subjected to 12 different thermal scenarios, taking care of possible variations in the thermo–physical properties of the materials in use (air cavity included). The typical FE result is presented in Figure 16 (a detailed example for January temperatures), where the sensitivity of the $f_{Rsi,FE}$ values from Equation (7) is shown. For evidence, we present the effects of the local details of the CLT–glass façade element on the thermo–physical performance indicators, with maximum variations close to the CLT frame and in the region of the corner joints. Even under various temperature boundaries (see Table 9) stable performance was also observed for different months of the year, further enforcing the potential of the façade solution, as well as the need for more extended studies.

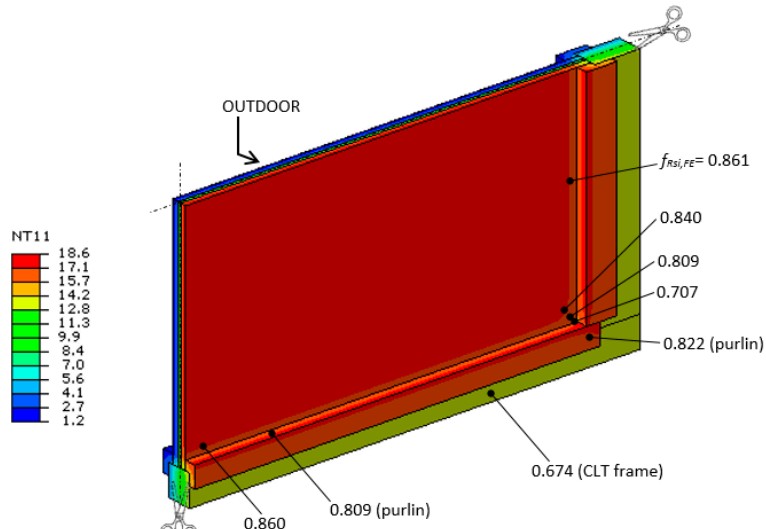

**Figure 16.** Numerical calculation of the temperature factor $f_{Rsi,FE}$ for the CLT–glass façade element (ABAQUS, with legend values in °C, with the calculation example referring to January temperatures).

**Table 9.** Numerical prediction of the critical temperature factor for the hybrid CLT–glass façade element (ABAQUS).

| | | Month | | | | | | | | | | | |
|---|---|---|---|---|---|---|---|---|---|---|---|---|---|
| | | Jan | Feb | Mar | Apr | May | Jun | Jul | Aug | Sep | Oct | Nov | Dec |
| Glass (minimum) | $T_{si,FE}$ (°C) | 14.4 | 15.0 | 16.2 | 17.6 | 19.1 | - | - | - | 18.9 | 17.5 | 18.1 | 14.6 |
| | $f_{Rsi,FE}$ | 0.706 | 0.708 | 0.708 | 0.708 | 0.709 | - | - | - | 0.708 | 0.709 | 0.859 | 0.705 |
| Glass (edge average) | $T_{si,FE}$ (°C) | 16.4 | 16.7 | 17.5 | 18.4 | 19.4 | - | - | - | 19.3 | 18.4 | 18.8 | 16.4 |
| | $f_{Rsi,FE}$ | 0.809 | 0.809 | 0.808 | 0.809 | 0.810 | - | - | - | 0.810 | 0.814 | 0.908 | 0.809 |
| CLT frame | $T_{si,FE}$ (°C) | 13.8 | 14.5 | 15.9 | 17.4 | 18.9 | - | - | - | 18.7 | 17.1 | 17.9 | 13.9 |
| | $f_{Rsi,FE}$ | 0.674 | 0.678 | 0.676 | 0.682 | 0.675 | - | - | - | 0.673 | 0.663 | 0.843 | 0.674 |

## 7. Conclusions

In this paper, a hybrid structural element composed of Cross-Laminated timber (CLT) and laminated glass was analyzed. The development of the design concept for this element started in 2006, when the first study was carried out within the Croatian project "Composite structural systems timber-structural glass and timber-steel" (coordinator prof. V. Rajčić, with funding from the Ministry of Education and Science) and later within the project "Seismic resistance of composite timber-structural glass structural systems with the optimal level of energy dissipation" (2010–2012, Croatia-North Macedonia bilateral agreement, coordinators prof. V. Rajčić and prof. L. Krstevska). Further developments were then made within the project "VETROLIGNUM—Multipurpose structural panel" (Croatian Science Foundation, coordinator prof. V. Rajčić).

Unlike traditional solutions used for façades (i.e., cellular frame-supported glass panels), this design concept is based on the avoidance of stress concentrations at the CLT and laminated glass joints due to the use of mechanical fasteners or adhesives. The joint was designed as a contact joint over friction surfaces. The hybrid element showed excellent performance under static, dynamic, and earthquake loads and was designed to have good structural performance. The primary purpose of these large panels is to be used as façade elements, but several other characteristics and performance parameters should be addressed and possibly optimized (i.e., energy efficiency, water-tightness, airtightness, optimal thermal characteristics, durability, and required lightening comfort).

The focus of this paper, accordingly, was to preliminary evaluate the thermal performance of the hybrid CLT–glass prototype under ordinary operational conditions. The analysis efforts were supported by continuous registrations and thermo graphical acquisitions from a Live-Lab mock-up building that was installed at the University of Zagreb. The expected thermal performance was first analyzed based on the expected behavior of the panel prototype under typical climate configurations in the Croatian region by using the Live-Lab records. The thermal and energy performance indicators for a full-size building were thus assessed via a computationally efficient but professionally-vetted commercial software code (EnCert-HR).

A further refined Finite Element numerical model (ABAQUS) was also developed to locally evaluate the typical thermal response of a full-size CLT–glass façade component. The numerical model, as shown, proved promising performance for the hybrid panel in line with European and national requirements. Critical temperature distributions were again observed in the region of the corner joints, which is in agreement with the Live-Lab acquisitions.

In subsequent stages of the ongoing study, most investigations will be focused on improving the design details in the contact areas between different building components and materials, as well as investigating new sealing products and various options for adaptive solutions (such as shading systems).

**Author Contributions:** Conceptualization, V.R. and C.B.; data curation, N.P., C.B., and J.B.; formal analysis, N.P. and C.B.; funding acquisition, V.R.; investigation, V.R., N.P., and J.B.; methodology, V.R. and C.B.; project administration, V.R., N.P., and J.B.; resources, V.R.; software, N.P. and C.B.; supervision, V.R., C.B., and R.Ž.; validation, V.R., N.P., C.B., J.B., and R.Ž.; visualization, C.B. and R.Ž.; writing—original draft, V.R., N.P., and C.B. All authors have read and agreed to the published version of the manuscript.

**Funding:** This research was funded by The Croatian Science Foundation (Project no. IP-2016-06-3811 VETROLIGNUM—"Prototype of multipurpose composite timber-load bearing glass panel", coordinator Prof. Vlatka Rajčić, University of Zagreb, Croatia).

**Conflicts of Interest:** The authors declare no conflicts of interest. The funders had no role in the design of the study; in the collection, analyses, or interpretation of data; in the writing of the manuscript, or in the decision to publish the results.

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
