# Peer review of "Thermal and Energy-Efficiency Assessment of Hybrid CLT–glass Façade Elements"

_applsci, doi:10.3390/app10093071_

Round 1

Reviewer 1 Report

The paper presents an experimental investigation of a hybrid CLT-glass façade element.

The article starts with an introduction about the general requirements of buildings façades (e.g. energy efficiency, waterproofness, airtightness), reporting some references to recent researches.

In particular, the article presents the thermal characteristic of a component developed within the framework of an international project. The analyzed technology is a hybrid element presented as a system with excellent structural performance which is non optimized, at the actual phase of the research, for the energy performances.

In the article the mockup, material property, boundary conditions for the thermal analysis and the instrument for field experimental measurements are described. The presented results try to characterize the thermal behavior of the technology.

General observations:

A complete linguistic revision is required.

Authors are invited to take more care about the template and to revised the manuscript in order to correct mistakes all along the paper.

Just to report some examples:

- All the table have to be revised, in particular Table 3 and 4.

- Caption of figure or table have to immediately follow/come first the Figure/Table (e.g. Figure 6 and 11, Table 8)

- Sentences to be rephrased: #29, #498-#503,

The manuscript does not deal with the structural-seismic performance of CLT or hybrid timber structure. Please avoid general non discussed (or not referred, avoid self-citations) sentences like #67-#74, #508-#510

2. Thermal and energy efficiency assessment of facade components

#123-130 This part references to different parameters cited in EN ISO 13788. Please clearly indicate that this part has been taken from the standard.

#132-143 Could be useful to have some references to the problems listed by the authors.

#145-152 This part looks like unlinked to the rest of the paragraph: it is not clear why the authors want to talk about the climate changes in general way. Please insert some relevant citation.

5.Full-size numerical analysis of the Live-Lab facility

# 320 According to Figure 4 the mock-up looks to be near another buildings: is this aspect considered in the analysis?

# 341 In the Tables there’s no unit for the properties

# 358 Please check the values of ““Vapour pressure in space’ and ‘Saturated vapour pressure’.

Frsi (1) is related to Tsi, Toul e Tint, therefore it could be useful add these parameters in Table 7

6. Thermal numerical analyses at the CLT-glass component level

# 459-461 In order to have a simpler and direct comprehension of Figure 13b it is suggested to specify into the caption of Figure 13 the meaning of CP1, CP2,CP3.

# 491. Please give and in-depth explanation about the high difference between the ???? calculated in Section 2 and the ???? reported at (6): it could be useful to report the value of the different Temperatures used in formula (6). Furthermore, is required to explicitly compare and discuss the ???? obtained from numerical simulation and ???? given by formula (6).

7. Conclusion

# 529 Please report short consideration on the difference between the ???? numerically assessed and the one calculated on the experimental data

As discussed in #491-494 please report this statement as an important results and please underline that further investigation are required.

Author Response

REVIEWER #1

Comments and Suggestions for Authors

The paper presents an experimental investigation of a hybrid CLT-glass façade element.

The article starts with an introduction about the general requirements of buildings façades (e.g. energy efficiency, waterproofness, airtightness), reporting some references to recent researches.

In particular, the article presents the thermal characteristic of a component developed within the framework of an international project. The analyzed technology is a hybrid element presented as a system with excellent structural performance which is non optimized, at the actual phase of the research, for the energy performances.

In the article the mockup, material property, boundary conditions for the thermal analysis and the instrument for field experimental measurements are described. The presented results try to characterize the thermal behavior of the technology.

General observations:

A complete linguistic revision is required.

The whole document is now fully revised and improved in language.

Authors are invited to take more care about the template and to revised the manuscript in order to correct mistakes all along the paper. Just to report some examples:

  • All the table have to be revised, in particular Table 3 and 4.
  • Caption of figure or table have to immediately follow/come first the Figure/Table (e.g. Figure 6 and 11, Table 8)
  • The authors agree. Unfortunately, some of the captions wrongly moved do to the file conversion from the online system. This aspect is now improved in the revised paper.
  • Sentences to be rephrased: #29, #498-#503,
  • All these sentences are corrected.

The manuscript does not deal with the structural-seismic performance of CLT or hybrid timber structure. Please avoid general non discussed (or not referred, avoid self-citations) sentences like #67-#74, #508-#510

A: The authors agree that this paper is not focused on the seismic performance of hybrid timber structures. However, the key aspect is that this paper explores for thermal assessment purposes the same CLT-glass façade system that has been proposed for seismic design in previous documents. Thus, a direct reference cannot be omitted. In order to avoid misleading general comments in the paper.

  1. Thermal and energy efficiency assessment of facade components

#123-130 This part references to different parameters cited in EN ISO 13788. Please clearly indicate that this part has been taken from the standard.

Done.

#132-143 Could be useful to have some references to the problems listed by the authors.

Done. See #627 - #644

#145-152 This part looks like unlinked to the rest of the paragraph: it is not clear why the authors want to talk about the climate changes in general way. Please insert some relevant citation.

The mentioned paragraph is removed from the paper, based also on other review recommendations.

5.Full-size numerical analysis of the Live-Lab facility

# 320 According to Figure 4 the mock-up looks to be near another buildings: is this aspect considered in the analysis?

It is considered wit solar gains and loses.

# 341 In the Tables there’s no unit for the properties

Done. You can see it in #328, table 4.

# 358 Please check the values of ““Vapour pressure in space’ and ‘Saturated vapour pressure’.

Done, see #353, table 5.

Frsi (1) is related to Tsi, Toul e Tint, therefore it could be useful add these parameters in Table 7

Done, see #353, table 5.

  1. Thermal numerical analyses at the CLT-glass component level

# 459-461 In order to have a simpler and direct comprehension of Figure 13b it is suggested to specify into the caption of Figure 13 the meaning of CP1, CP2,CP3.

Done.

# 491. Please give and in-depth explanation about the high difference between the ???? calculated in Section 2 and the ???? reported at (6): it could be useful to report the value of the different Temperatures used in formula (6). Furthermore, is required to explicitly compare and discuss the ???? obtained from numerical simulation and ???? given by formula (6).

Some comments are now included in the paper, to clarify this scatter.

  1. Conclusion

# 529 Please report short consideration on the difference between the ???? numerically assessed and the one calculated on the experimental data

Done.

As discussed in #491-494 please report this statement as an important results and please underline that further investigation are required.

Done.

Reviewer 2 Report

The thermal and energy-efficiency assessment of novel hybrid CLT-glass facade elements is described in the presented paper. The paper contains a lot of issues that must be corrected.

In general, the paper is very poorly structured and should be completely rewritten. The authors jump from one topic to another one, while each topic is only slightly tackled. It is not clearly explained how the information is relevant to the main topic of the paper. The aim of the paper is nicely stated in the abstract and that's all. The rest of the paper, except chapters 5 and 6, is quite messy and contains a lot of useless information. Please, try to remove all these euphemisms. The beginning of the article is too long.

Line 29: “The numerical results show that the even the use of a bare …“. Please delete the article before even.  

Line 37: “…, firstly beams.” What’s the meaning of these words, it is quite difficult to understand.

Line 39: Please use conjunction instead of the symbol /.

Line 46: There is no link between the text and the presented Figure.

Line 55 – 62: The panel described in this paragraph is the same panel as the one that was described by references 7-9? I assume that yes, and therefore, there should be a reference number in the description of Figure 2. Moreover, it is very difficult to follow the presented description. The text is not very consistent and needs some changes. Also, it is said that: “laminated semi-tempered layers bonded by Ethylene Vinyl Acetate (EVA®) foils.”, or “the thickness of the Bridgestone EVASAFE adhesive bond was 1.6mm”. Please can you show the layer of the material in Figure 2, or at least to highlight the area where the material is used.

Line 69 – 81: This part of the article should be in my opinion mentioned before lines 52 - 64. The paragraph has to be changed; the content is quite useless since only commonly known information are presented.

Line 90: Please, can you name what type of specific performance limits must be satisfied.

Line 87 - 95: There are 3 paragraphs. The information described in each paragraph is the same. Please, try to rewrite these paragraphs, the current presentation is not very fortunate, e.g. what do you mean by more advanced calculations?

Line 97: “, among others, “ – Why? Is it necessary to use these to words? Moreover, the sentence is difficult to understand and should be rewritten. I think that it would be better to say something like this: In the presented paper, the EN ISO 13788 standard was followed. The standard is focused on description of the assessment of the hygrothermal performance of building components and building elements.

Line 106: “Another …”. This sentence should start a new paragraph.

Line 159: What is “forest climate”?

Line 162 – 163: Please, can you explain how a study (or data) proposed in 2003 underestimates the real climate that was witnessed in recent years? Moreover, data recorded within 3 years should not be considered as reliable. Long-term measurements are needed. Again, please, try to rewrite this paragraph. The only interesting information is described in lines 163-165. I would delete the 1st and 2nd paragraphs of this subchapter.

Line 188-189: ”Prefabricated panels of typical use for the construction of wooden houses were used, …”. What do you mean? It is difficult to understand what is meant by “typical use”. By panel, do you mean SIPs panel or prefabricated 2by4 system panel? The authors should really put more effort to clearly describe the system and material that was used.

Line 221-222: It is said that there was a meteorological station on the roof of the Live-Lab. The lab was constructed in 2018. However, in line 164 it is said that the meteorological data were recorded over a period of 3 years. It is the beginning of 2020. This does not make sense, please, can you explain what type of data was recorded and where.

Line 244: Why do you start with Figure 7? Shouldn't be Figure 6 mentioned before number 7?

Page 11: Table 1 and Table 2, this information should be mentioned before chapter 4.

The content of chapter 5 and chapter 6 is the main of the paper, as stated in the abstract, however, it looks like a totally different article, there is no link between these chapters and the previously mentioned ones. Moreover, it is too evident that this part was probably written by a different person.

Author Response

REVIEWER #2

Comments and Suggestions for Authors

The thermal and energy-efficiency assessment of novel hybrid CLT-glass facade elements is described in the presented paper. The paper contains a lot of issues that must be corrected.

In general, the paper is very poorly structured and should be completely rewritten. The authors jump from one topic to another one, while each topic is only slightly tackled. It is not clearly explained how the information is relevant to the main topic of the paper. The aim of the paper is nicely stated in the abstract and that's all. The rest of the paper, except chapters 5 and 6, is quite messy and contains a lot of useless information. Please, try to remove all these euphemisms. The beginning of the article is too long.

Line 29: “The numerical results show that the even the use of a bare …“. Please delete the article before even.

Done.

Line 37: “…, firstly beams.” What’s the meaning of these words, it is quite difficult to understand.

The text is now revised and improved.

Line 39: Please use conjunction instead of the symbol /.

Done.

Line 46: There is no link between the text and the presented Figure.

A sentence is added to clarify the motivation of the figure.

Line 55 – 62: The panel described in this paragraph is the same panel as the one that was described by references 7-9? I assume that yes, and therefore, there should be a reference number in the description of Figure 2. Moreover, it is very difficult to follow the presented description. The text is not very consistent and needs some changes. Also, it is said that: “laminated semi-tempered layers bonded by Ethylene Vinyl Acetate (EVA®) foils.”, or “the thickness of the Bridgestone EVASAFE adhesive bond was 1.6mm”. Please can you show the layer of the material in Figure 2, or at least to highlight the area where the material is used.

As observed, the original text was inclusive of some repetitions and misleading descriptions. Given that the hybrid system is the same of refs. 7-9, the paper text is now improved.

Line 69 – 81: This part of the article should be in my opinion mentioned before lines 52 - 64. The paragraph has to be changed; the content is quite useless since only commonly known information are presented.

As suggested (and based also on comments from Reviewer #1), the mentioned text has been partly moved in advance to the current position, and also partly shortened.

Line 90: Please, can you name what type of specific performance limits must be satisfied.

Please notice that the mentioned sentence has been partly removed and rearranged, to avoid repetitions or misleading comments in the paper.

Line 87 - 95: There are 3 paragraphs. The information described in each paragraph is the same. Please, try to rewrite these paragraphs, the current presentation is not very fortunate, e.g. what do you mean by more advanced calculations?

The mentioned sentences are now rewritten and improved in content / form, based also on other review suggestions.

Line 97: “, among others, “ – Why? Is it necessary to use these to words? Moreover, the sentence is difficult to understand and should be rewritten. I think that it would be better to say something like this: In the presented paper, the EN ISO 13788 standard was followed. The standard is focused on description of the assessment of the hygrothermal performance of building components and building elements.

The sentence is now rearranged and improved in its content.

Line 106: “Another …”. This sentence should start a new paragraph.

Revised.

Line 159: What is “forest climate”?

Based also on further review comments, the mentioned paragraph has been removed from the revised paper.

Line 162 – 163: Please, can you explain how a study (or data) proposed in 2003 underestimates the real climate that was witnessed in recent years? Moreover, data recorded within 3 years should not be considered as reliable. Long-term measurements are needed. Again, please, try to rewrite this paragraph. The only interesting information is described in lines 163-165. I would delete the 1st and 2nd paragraphs of this subchapter.

It is generaly recognized that mean temperatures are worldwide progressively increasing. This was the motivation of the original comment. However, most of the chapter (and figure 3a) has been removed from the revised paper.

Line 188-189: ”Prefabricated panels of typical use for the construction of wooden houses were used, …”. What do you mean? It is difficult to understand what is meant by “typical use”. By panel, do you mean SIPs panel or prefabricated 2by4 system panel? The authors should really put more effort to clearly describe the system and material that was used.

Please check #178 - # 184 and figure 5.

Line 221-222: It is said that there was a meteorological station on the roof of the Live-Lab. The lab was constructed in 2018. However, in line 164 it is said that the meteorological data were recorded over a period of 3 years. It is the beginning of 2020. This does not make sense, please, can you explain what type of data was recorded and where.

Data were recorded from 2018. Now the text is revised.

Line 244: Why do you start with Figure 7? Shouldn't be Figure 6 mentioned before number 7?

Yes, this detail is now corrected.

Page 11: Table 1 and Table 2, this information should be mentioned before chapter 4.

Done.

The content of chapter 5 and chapter 6 is the main of the paper, as stated in the abstract, however, it looks like a totally different article, there is no link between these chapters and the previously mentioned ones. Moreover, it is too evident that this part was probably written by a different person.

In accordance with the comments and revision, this section has now been revised and linked.

Round 2

Reviewer 1 Report

The authors have addressed all the comments. The paper is now ready to be published.

Author Response

Thank you for the concluding that all the comments have been addressed properly.

Reviewer 2 Report

Line 45: ... according to 5 - this way of referring to a particular design or implementation does not make any sense. Moreover, are the authors really considering the reader to go to find out ref 5 to find that information?

Author Response

Dear revier, it is obligatory to cite the source of the figure so we think that the reference to 5 should stay as it is.